# Medium-Term Effects of Sprinkler Irrigation Combined with a Single Compost Application on Water and Rice Productivity and Food Safety

**DOI:** 10.3390/plants12030456

**Published:** 2023-01-19

**Authors:** David Peña, Carmen Martín, Damián Fernández-Rodríguez, Jaime Terrón-Sánchez, Luis Andrés Vicente, Ángel Albarrán, Jose Manuel Rato-Nunes, Antonio López-Piñeiro

**Affiliations:** 1Área de Edafología y Química Agrícola, Escuela de Ingenierías Agrarias—IACYS, Universidad de Extremadura, Ctra de Cáceres, 06071 Badajoz, Spain; 2Área de Edafología y Química Agrícola, Facultad de Ciencias—IACYS, Universidad de Extremadura, Avda de Elvas s/n, 06071 Badajoz, Spain; 3Área de Producción Vegetal, Escuela de Ingenierías Agrarias—IACYS, Universidad de Extremadura, Ctra de Cáceres, 06071 Badajoz, Spain; 4Instituto Politécnico de Portalegre, Escola Superior Agraria de Elvas, 7350-092 Elvas, Portugal

**Keywords:** direct seeding, efficient irrigation method, organic compost, rice productivity

## Abstract

Traditional rice (*Oryza sativa* L.) management (tillage and flooding) is unsustainable due to soil degradation and the large amount of irrigation water used, an issue which is exacerbated in the Mediterranean region. Therefore, there is a need to explore rice management strategies in order to improve water-use efficiency and ensure its sustainability. Thus, field experiments were conducted to determine the medium-term effects of different irrigation and tillage methods combined with a single compost application on water and rice productivity, as well as food safety in a semiarid Mediterranean region. The management systems evaluated were: sprinkler irrigation in combination with no-tillage (SNT), sprinkler irrigation in combination with conventional tillage (ST), which were implemented in 2015, and flooding irrigation in combination with conventional tillage (FT), and their homologues (SNT-C, ST-C, and FT-C) with single compost application in 2015. In reference to rice grain yield, the highest values were observed under ST treatment with 10 307 and 11 625 kg ha^−1^ in 2018 and 2019 respectively; whereas between FT and SNT there were no significant differences, with 8 140 kg ha^−1^ as mean value through the study. Nevertheless, sprinkler irrigation allowed saving 55% of the total amount of water applied in reference to flooding irrigation. Furthermore, the highest arsenic concentration in grains was found under FT but it decreased with compost application (FT-C) and especially with sprinkler irrigation, regardless of tillage management systems. However, sprinkler irrigation favors the cadmium uptake by plants, although this process was reduced under SNT in reference to ST, and especially under amended compost treatments. Therefore, our results suggested that a combination of sprinkler irrigation and compost application, regardless of the tillage system, could be an excellent strategy for rice management for the Mediterranean environment in terms of water and crop productivity as well as food safety.

## 1. Introduction

One of the most relevant challenges to agriculture in the immediate future is the management of soil and water resources that are increasingly degraded and scarce [1]. A clear example of that is rice, which is a crucial crop to global food safety, but traditionally cultivated through unsustainable practices which include intensive tillage techniques and flooding irrigation [2]. In this sense, intensive tillage operations aggravate the problem through the accelerating of soil organic matter, the disruption of soil structure, and the decline of biological activity, thus compromising the yields and sustainability of crops [3]. In order to improve these environmental impacts, conservation tillage practices need to be implemented [4], such as “direct seeding” through the total elimination of the tillage operations. It is especially important in the Mediterranean area, whose soils are characterized by low organic matter content due to inadequate soil management [5]. However, although no-tillage management could be used to improve soil conservation, its impact on rice growth is not fully understood or still controversial [6]. In fact, Sánchez-Llerena et al. [7] conducted a field experiment under Mediterranean conditions, indicating that no-tillage management improves soil properties, leading to an increase in rice yield. Nevertheless, Singh et al. [8] observed that rice yield was significantly lower under no-tillage than under tillage management due to differences in weed infestation between both managements. Generally, the lack of soil disturbance under no-tillage conditions favors the soil properties through soil organic matter content, being the time of implementation a crucial factor on crop yields. Thus, whereas significant decreases have been observed in the short-term implementation of no-tillage management [9] increases in rice yields had been found under long-term implementation [10]. These differences could be caused by soil organic matter content which at long-term is higher than short-term implementation [11]. In addition, several studies had been reported that the application of organic amendments could increase rice yield by increasing the storage of soil organic matter and soil nutrient content [12]. However, due to the numerous variables involved such as soil properties, type of organic amendments, and its rate, as well as climatic conditions, determining the impacts of organic amendments on rice yield is quite difficult [13]. Besides, the application of organic amendments improves soil water retention, and thereby, plant available water [14], contributing to increasing water productivity. That is particularly crucial in rice crops whose water consumption under the traditional system may reach up to 5 000 L to produce 1 kg of rice [15]. This situation is environmentally unsustainable, especially in Mediterranean regions where water resources are limited. However, rice growing, which is mainly located in the poorest regions of the European Union, plays quite an important role in the socio-economic developments of these regions. In this sense, different water-saving methods have been developed to improve water-use efficiency in rice crops [16]. In fact, Spanu et al. [17] in a 2-year field study under Mediterranean climatic conditions and for 26 rice genotypes found that average yields of rice irrigated by flooding and sprinkler were never statistically different from each other. Therefore, the transition from flooding to sprinkler rice irrigation could lead along with similar yields important environmental advantages such as halved water requirements, the avoidance of soil leveling and the use of specific agricultural machinery, and reduce the number and intensity of treatments against weeds [7,17,18]. However, other studies showed that aerobic rice can lead to decreases in rice yield in reference to flooded conditions [19,20]. In fact, Bozkurt-Colak et al. [21], in a 2-year field experiment in the Mediterranean Region, indicated that the implementation of aerobic rice could lead to significant reductions of up 40% in the rice yield in reference to the permanent flooding irrigation system. Therefore, identifying the factors that determine the differences in yield levels between aerobic and flooded management is a crucial step towards the sustainable productivity of rice. Furthermore, studies determining the medium- and long-term effects of rice management alternatives on yield parameters are needed in order to validate their effectiveness because most research explores the short-term effects [22]. 

Another important issue associated with flooded conditions is the high level of metalloids and heavy metals in rice grain, which leads to serious health problems [23]. In particular, compared to other cereals such as maize, wheat, or barley; rice accumulates a much higher content of arsenic (As) and cadmium (Cd) in the grains, causing a global environmental health concern [24]. Several studies have indicated that water management cause changes in physicochemical soil properties, which may affect the As and Cd uptake by rice plants [25]. However, the trends found in studies evaluating the effects of rice water management on metal accumulation in grain have often been contradictory. Thus, whereas different studies have found that the transition from flooding to aerobic irrigation in rice growing decreased considerably the As concentration but enhanced Cd transfer to grain [26,27], Spanu et al. [17,28] showed a decrease in As and Cd grain concentration under a sprinkler in comparison to flooding irrigation. Therefore, further research is needed to develop an effective rice management in order to reduce the uptake of toxic elements. In this sense, previous researchers have found that the application of organic amendments could reduce Cd bioavailability [29]. However, Bai et al. [30] observed that straw application increased the uptake of Cd by rice plants, showing again that processes of metal translocation in amended soils are quite complex [31]. 

The olive oil industry is very important for the socioeconomic development of Mediterranean countries, where more than 90% of the world’s olive oil is produced. However, these countries are facing serious environmental issues regarding the seasonal generation of large amounts of waste [32]. Only in Spain over five million tonnes of two-phase olive mill waste are produced annually, and therefore it is necessary to find useful practices for its disposal. This waste contains up to 85% of organic matter and its use as organic amendment may therefore lead to an excellent alternative for its management, thereby achieving one of the European Union about Circular Economic Strategy essential aims: the wastes should be reused [33]. However, two-phase olive mill waste must be correctly managed to avoid serious environmental effects such as decreases in soil quality and contamination of atmospheric and pollution of aquatic ecosystems [34], or even toxicity problems for the crops. In this sense, several investigations have proposed that composting is a suitable method for two-phase olive mill waste valorization [35].

In spite of the unsustainability of rice crops under traditional management, particularly in Mediterranean regions with semi-arid climate conditions, there is little research about the impact of alternative managements on rice productivity, as well as food safety. Besides, to the best of our knowledge, there have as yet not been studies that determining the medium-term effects of composted two-phase olive mill waste application under different irrigation (flooding and sprinkler) and tillage (conventional tillage and direct seeding) methods, and clearly such information is required to determine the viability. Thus, field experiments were conducted to determine the medium-term effects of different irrigation and tillage methods combined with a single compost application on water and rice productivity, as well as food safety in a semiarid Mediterranean region. In addition, the present results have been closely compared with previous studies where the short-term effects of these methods on water and rice productivity [36] and food safety [29] were analyzed in order to test the longevity of these alternatives, a basic step for their implementation as sustainable rice systems.

## 2. Materials and Methods

### 2.1. Study Site, Experimental Design and Field Management

The study was carried out under field conditions located in Gévora (38°55’ N; 6°57’ W altitude of 180 m a.s.l.), Southern Spain. The climate is Mediterranean, characterized by dry and hot summers and rainfall mainly concentrated in winter (with annual values < 480 mm). The soil is classified as Hydragic Anthrosol [37] whose texture is loam, with 20.8% clay, 28.9% silt, and 50.3% sand. This study site had been dedicated to rice (*Oriza sativa* L.) monocropping for a long time under traditional management (flooding and deep ploughing). Thus, after the rice harvest in December 2014, the field was divided into 180-m^2^ experimental blocks and was subjected to six different management systems. These systems were: rice-growing under sprinkler irrigation and no tillage (direct seeding) without (SNT) and with the application of compost (SNT-C); rice-growing under sprinkler irrigation and conventional tillage without (ST) and with the application of compost (ST-C); and rice-growing under flooding irrigation and conventional tillage without (FT) and with application of compost (FT-C). SNT and ST were implemented in order to determine the effects of tillage on water and rice productivity, as well as food safety under a water saving method (sprinkler irrigation). All treatments were carried out in triplicate, therefore, the experimental field showed eighteen blocks. A single application of compost in April 2015 was performed at rate of 80 Mg ha^−1^. The main properties of soils and compost are presented in Appendix A. Hence, in order to determine the medium-term effects of these managements on agronomic rice parameters and metals accumulation in rice grains the data shown in the present study correspond with two rice cropping cycles, 2018 and 2019, four and five years after their implementation, respectively. The data of temperature, rainfall, and rice evapotranspiration (*ET_C_*) registered at the field location during 2018 and 2019 are shown in Appendix A. Briefly, rice, as the sole crop, was sown at a dosage of 160 kg ha^−1^ seeds of *Oryza Sativa* L. in early May and harvested at the end of September each year. Every year, all treatments received three fertilizer applications, one with (9-18-27) complex fertilizer as basal dressing before sowing, and two applications of urea as N fertilizer during the rice growing cycle. Rice was irrigated with Guadiana river water through a sprinkler system in ST, ST-C, SNT, and SNT-C giving total coverage, and then via flooding in FT and FT-C. The water applied was monitored by water flow-meters, being the consumption of water under sprinklers far lower than the flooding system. Hence, the water supply in sprinkler irrigation treatments was 8 607 m^3^ ha^−1^ and 8 879 m^3^ ha^−1^ for 2018 and 2019, respectively; whereas in flooding irrigation treatments was 16 275 and 15 350 m^3^ ha^−1^ for 2018 and 2019, respectively. Pre-emergence weed control was performed using 1.5 kg ha^−1^ of Pendimethalin for the rice irrigated by sprinkler, whereas the rice irrigated with continuous flooding was treated with 360 g ha^−1^ of Clomazone, and a mixture of Imazamox (70 g ha^−1^) and 2- methyl-4-chlorophenoxyacetic acid (500 g ha^−1^) for postemergence weed control.

### 2.2. Data Collection

#### 2.2.1. Weed Control Efficiency

Weed samples were collected from 30 × 30 cm quadrats per plot (sampling unit) in order to determine the impact of the different treatments on weed control efficiency (WCE), which was calculated as: WCE = (DWC − DWT)/DWC (1), where DWC is dry weight of weeds in non-treated plots and DWT is the dry weight of weeds in treated plots [38]. 

#### 2.2.2. Agronomic Parameters

In each plot, agronomic parameters were analyzed from a 2 m^2^ area (6 m^2^ for each treatment). In addition, the standard moisture of 0.14 g H_2_O g^−1^ fresh weight was adjusted to production parameters. The ratio between sprouting seeds and total seeds sown is used to calculate the germination index (GI). The ripening index percentage (RI) was calculated as the percent ratio of the filled grain number and total grain number. Grain yield (Y) was determined as the direct weight of all filled grains per panicle collected in the trial area. Water productivity (WP) was defined as the ratio between Y and the applied amount of irrigation water. 

#### 2.2.3. Arsenic and Cadmium in Soil and Rice Grain

After the harvest, four subsamples of soil from 0–20 cm depth were taken for each plot by a manual auger, air-dried, and milled in an agata mortar to obtain particles with a lower grain size (<0.2 mm). Likewise, a significant sample of rice grain from each plot was dried at 60 °C to constant weight and then dehusked, milled, and sieved to 0.2 mm. In these samples were determined the total arsenic (As) and cadmium (Cd) concentrations by atomic emission spectroscopy as described by Alvarenga et al. [29]. In addition, in order to enhance the knowledge about the effect of different system managements on food safety, arsenic speciation in the rice grain was also determined as described by Alvarenga et al. [29]. 

### 2.3. Statistical Analyses

The SPSS software package (22.0) was used to perform the statistical analyses. The data were subjected to a one-way ANOVA, after verifying the normality and homogeneity of variances, to determine the significance of treatment and year. Duncan’s test was used for multiple comparisons. In order to find significant correlations between the results Pearson’s correlations were carried out. Furthermore, the data were also subjected to a two-way ANOVA to determine the significance of the interaction (Year X Treatment). Statistical significant differences at the 0.05, 0.01, and 0.001 level of probability were indicated by *, **, and ***, respectively. 

## 3. Results and Discussion

### 3.1. Weed Control Efficiency

The effects of management systems on herbicide efficacy are presented in Figure 1. For both years, in the original managements, the lowest values were found under FT treatment, with values of 16.7 and 23.2% of herbicides efficacy in 2018 and 2019 (Figure 1). However, under sprinkler irrigation, the application of herbicides was an effective strategy for weed control, especially under tillage methods, whose values were over 80% in both years of the study (Figure 1). Therefore, our results suggest that the permanent flooding irrigation system did not ensure an effective control over weeds in rice crops. Similarly, different authors (e.g., [39]) had indicated that the continuous use of permanent flooding systems under rice monoculture caused that weed species to be well-adapted to this condition. Furthermore, there are recent researches that indicated different rice management systems can modify the behavior of herbicides. In fact, there are studies that had been reported a faster dissipation of herbicides in flooded than non-flooded soil conditions, for different herbicides widely applied in rice crops such as bispyribac-sodium, clomazone, MCPA [40,41,42]. Therefore, the short persistence under flooding irrigation could lead to a lower herbicide efficacy. Similar to found Peña et al. [36] for short-term effects, the medium-term effects of compost on herbicides efficacy in reference to original management were not significant, showing a similar trend between them (Figure 1), suggesting that the water management and tillage operations are the principal factors on weed control for rice crop.

### 3.2. Agronomic Parameters

The effects of management systems on rice yield components are shown in Table 1. Regardless, of the tillage system, the sprinkler treatments showed values of germination index (GI) significantly higher than flooding treatments, in both year of the study (Table 1). Similar results has been found by Chamara et al. [43] who indicated that flooding irrigation reduces seed rice germination and crop stand but helps in weed control. Furthermore, regardless of the treatment, the middle-term effects of compost on GI were not significantly (Table 1), even though compost increased the pH of soils (Appendix A), property that showed a significant and positive correlation with GI (r = 0.551 **). Similar findings were observed by Wijayanto et al. [44] who indicated the importance of soil pH on germination and growth of rice due to its effect on level of toxicity and/or deficiency to certain minerals.

In reference to the ripening index (RI), whereas in 2018 there were no significant differences between treatments, with an average value of 88.2% (Table 1), in 2019 the highest value was observed under ST treatment (82.3%, Table 1), without find significant differences between SNT and FT (73.2% and 74.7%, respectively, Table 1). These results demonstrate that the potential risk of water stress in rice under aerobic rice, which has been observed by different authors [45] disappears at least in the medium term. In fact, under tillage operations, after 5 years of implementation, the sprinkler irrigation (ST) showed higher values of RI than flooding (FT) (Table 1), however, under the same treatments Peña et al. [36] found opposite tendency (higher RI in FT than ST) but for short-term effect, indicating the importance of evaluated the effects beyond short-term. The effects of compost were not significant under any management system (Table 1), probably because the amounts of water applied were meet the water requirements of rice crops (*ETc*). Furthermore, it is important to note the significant decreases found in 2019 in reference to 2018 in all treatments (except ST-C), a result that could be due to increases in EC values (Appendix A), as shown by the significant and negative correlation between RI and EC (r = −0.674 **). Likewise, other authors (e.g., [46]) indicated that rice is a salt-sensitive crop because its growth and yield components were significantly reduced by salinity stress. Besides, another possible reason to explain the trend of RI values was the herbicide’s efficacy, in fact, a significant and positive correlation was found between both (r = 0.534 **). Similar results were indicated by Korres et al. [47] who showed that salinity and weed are able to act synergically, increasing the negative effects on rice growth, suggesting that future research should be focused on these issues.

The parameter of grain yield (Y) was significantly affected by the treatments (Table 1). Thus, Y values were greater in ST than SNT by factors of 1.12 and 1.64 for 2018 and 2019, respectively, and in ST than FT by factors of 1.24 and 1.47 for 2018 and 2019 (Table 1). Therefore, the medium-term effects of the transition from flooding to sprinkler irrigation on Y levels were positive, especially under tillage conditions. One explanation for these results could be the increases in pH values observed in sprinkler irrigation treatments in reference to flooding treatments (Appendix A). Indeed, Y was significant and positively correlated with pH (r = 0.361 *) indicating that pH could be an important factor in rice production, probably due to an improvement in nutrient availability [48]. Besides, a significant and positive correlation was also observed between Y and GI (r = 0.447 **) and Y and RI (r = 0.331 *), indicating that those management systems which improved rice growth also lead to grain yield increases. Furthermore, the trend observed in Y values could be to explain by differences observed in the herbicide’s efficacy between the treatments. In fact, Y values had a significant positive correlation with herbicide efficacy (r = 0.433 **) suggesting that is very important to achieve optimum weed control in rice crops in order to ensure their agronomic and economic viability. These are coherent with several studies that reported that weed competition is one of the major biotic issues for rice production (e.g., [39,49]). Despite there were no significant differences between years for any management system (the factor year was not significant, Table 1), is important to note that whereas under flooding irrigation the Y values were stagnant at around 8 000 kg ha^−1^, under sprinkler irrigation the variations depend on the tillage operations. Thus, under SNT the Y values decreased from 9 226 to 7 070 kg ha^−1^ whereas under ST the Y values increased from 10 307 to 11 625 kg ha^−1^ throughout the study period (Table 1). Furthermore, the values found under ST at medium-term effect were higher than those observed by Peña et al. [36] for the short-term, indicating that the productive capacity of this management increases over time. Similar results were observed by Madhukar et al. [50] who suggested strategies such as the conservation of soil and water resources, as well as the rotation of crops, in order to prevent rice yield stagnation under flooding irrigation. The differences found in the trend of Y values between treatments under sprinkler irrigation (SNT and ST) could be explained by salinity stress, since rice is a salt-sensitive crop, because under ST the mean value of EC was 2.00 dS m^−1^, without significant differences throughout the study period, however, under SNT the values of EC increasing from 1.45 to 5.97 dS m^−1^ (Appendix A). In reference to the effect of compost is important to emphasize the increase observed in Y values under SNT-C in reference to SNT for the last year of the study (8 436 kg ha^−1^ and 7 070 kg ha^−1^, respectively), suggesting that residual effects of compost application could be an interesting alternative to mitigate salinity stress in rice crop. Likewise, increases in resistance to salinity stress were reported in rice crops after organic amendment application [51,52]. In fact, in crops such as tomato and wheat, the application of organic amendment increased stomatal conductance and stomatal density under salinity conditions, as well as enhanced moisture content and sodium binding property in the soil, which reduced the salinity stress [53]. 

Our results showed a clear trend for water productivity (WP) values, with the highest values under ST and ST-C treatments and the lowest under FT and FT-C throughout the study (Table 1). Thus, after five years of management system implementation, WP values were significantly greater by factors of 1.65 and 2.54 in ST than in SNT and FT, respectively, suggesting that flooding could be not effective irrigation system for rice crops. These are in line with different studies, which showed that values of WP increased under different water-saving rice management such as alternate wetting and drying irrigation [54], aerobic rice system [36], and drip irrigation [55], demonstrating the sustainability of these management systems, especially in areas characterized by scarcity of water resources. The effects of compost application on WP were not significant in reference to the original treatments (Table 1). However, WP was positively and significantly correlated with Β-glucosidase and Urease (Appendix A) (r = 0.431 ** and r = 0.378 *), respectively; indicating that the rice managements used to enhance the soil microbial activity could also be useful to maximize the productivity of water applied in this crop [56].

### 3.3. Arsenic and Cadmium in Soil and Rice Grain

The effects of management systems on total As and Cd concentrations in soil (A) and rice grain (B) are presented in Figure 2. The values of total As concentrations in soils ranged from 3.99 to 6.17 mg kg^−1^ throughout the study period, without significant differences between the management system and between years. These values were close to those observed by Kabata-Pendias [57], who found a mean value of 6.83 mg kg^−1^, but lower than those observed by Rokonuzzaman et al. [58]; who found values of total As in soil ranged between 11.1 and 22.6 mg kg^−1^, and 101.6 mg kg^−1^ observed by Zhang et al. [59] in a paddy soil around a mine in China. Same as for As, the values of total Cd concentration in soils ranged from 0.190 and 0.275 mg kg^−1^ throughout the study period, without significant differences between the management system and between years (Figure 2A). This range of concentrations can be considered normal but is low when compared with values of paddy soil collected from near the mining area. In fact, in a review carried out by Carrijo et al. [60] reported values of total Cd from 0.009 to 6.4 mg kg^−1^ for paddy soil. Therefore, according to the values of As and Cd concentrations, our soils are suitable for rice production. Furthermore, the compost used had a total As concentration of 4 mg kg^−1^ and total Cd concentration of 0.148 mg kg^−1^ (Appendix A), therefore complied with the threshold values for As and Cd (40 mg inorganic As kg^−1^ and 2 mg kg^−1^, respectively) indicated in Regulation (EU) 2019/1009 of the European Parliament and of the Council.

Unlike in soil, the management systems had significant effects on total As and Cd concentration in rice grains (Figure 2B). Thus, whereas As concentrations were 0.253 and 0.274 mg kg^−1^ under FT in 2018 and 2019, respectively, and 0.178 and 0.195 mg kg^−1^ under FT-C in 2018 and 2019, respectively, when sprinkler irrigation was used, the values were below the quantification limit (<0.01 mg kg^−1^), regardless of treatment and year. Similar results had been indicated by several authors (e.g., [29,60]) who indicated that under flooding conditions As is more efficiently taken up by rice plants in reference to other water-saving irrigation methods, due to with permanent flooding arsenate is reduced to arsenite, which is less strongly adsorbed by soil ferric oxides and more efficiently taken up by rice plants [60]. However, the limit established by the European Union legislation in rice grain is 0.20 mg Kg^−1^ of inorganic As (As (III) + As (V)) due to its higher toxicity capacity than organic As. Therefore, the effects of management systems on inorganic As (iAs) in rice grain are presented in Figure 2C. Similar to total As, the irrigation methods play an important role in iAs concentration in rice grains. Thus, under sprinkler irrigation, the values of iAS were below the quantification limit (<0.04 mg kg^−1^), regardless of treatment and year, whereas under FT treatment 0.140 and 0.178 mg kg^−1^ were observed in 2018 and 2019, respectively (Figure 2C). Therefore, from a food safety point of view, these results suggested that rice production under aerobic irrigation, regardless of tillage methods, was a better option than flooding irrigation. Furthermore, is important to note that iAS values in rice grain decreased in FT-C in reference to FT, being these differences significant in the last year of the study (Figure 2C). Thus, the values of iAS decreased by factors of 1.55 and 1.89 in 2018 and 2019, respectively. These results could be very interesting due to the application of compost in its medium-term effects to achieve a reduction in iAs under permanent flooding irrigation. Besides, from an economic point of view for farmers, is also very important because the rice produced under FT-C can be destined for the production of food for infants and young children, whose limit established by the European Union legislation is 0.1 mg iAS kg^−1^. Similar results had been described by Sengupta et al. [61] who indicated that organic amendments reduced the As uptake by plants due to of organo-As chelates. However, Alvarenga et al. [29] showed no significant differences in iAS between FT and FT-C when the short-term effect was analyzed, probably due to changes in the properties of compost under five year aging at field conditions. 

The European Union legislation also limits the Cd content, aother important toxic element, in rice grain (0.2 mg kg^−1^). Unlike As, the values of Cd concentration were significant increased under sprinklers in reference to flooding irrigation, however, the limit of 0.2 mg kg^−1^ did not exceed by any management systems (Figure 2B). These results are coherent with several studies that also reported that water-saving irrigation reduces grain arsenic but enhances cadmium [24,62]. However, Spanu et al. [17,28] who studied the effect of the irrigation method on the bioaccumulation of toxic elements in rice grain, indicated that sprinkler irrigation could be interesting management in order to reduce the Cd concentration in rice grain, although these authors observed that this result could depend on the crop year. In this sense, the research carried out to determine the medium-term, as in the present study, or long-term effects take on even greater importance.

Furthermore, is important to highlight the differences found between SNT and ST treatments. Indeed, the values of Cd concentration in rice under SNT were 1.15 and 3.81 times lower than ST in 2018 and 2019, respectively (Figure 2B). Therefore, in the context of food security and after several years of implementation the grain produced under SNT is better than ST due to having a similar content in iAs but with a lower content in Cd. However, regarding short-term effects there were no difference between them [27]. Probably, these differences could be to explain by pH values, higher in SNT than ST (Appendix A), due to an increase in pH soil could lower the mobility of Cd through the forming and precipitation of Cd(OH)^−^ [63]. Similarly, under compost-amended treatments (SNT-C and ST-C) the Cd concentration significantly decreases in reference to their original management (Figure 2B). These effects could be explained by the formation of organo-Cd chelated, in particular, HA is able to offer absorption sites to bind Cd in soil [64], reducing the Cd uptake by plants. In fact, a significant and negative correlation was observed between HA and grain Cd concentrations (r = −0.375 *).

## 4. Conclusions

After five years of different sprinkler irrigation and tillage methods, combined with only one compost application, significant changes in water and rice crop productivity, and bioaccumulation of toxic elements in grain were observed. In fact, the use of sprinklers instead of flooding irrigation methods has allowed us to obtain a significant increase in yields, with halved water applied. Furthermore, the risk of arsenic accumulated in rice grain when flooding irrigation was used, decreased with residual compost effects, and was directly eliminated under sprinkler irrigation, regardless of the tillage methods. Nevertheless, the accumulation of cadmium in rice grains was favored in sprinkler irrigation conditions, especially under tillage management; although this risk could be reduced with the application of compost. Therefore, under semiarid Mediterranean conditions, the transition from flooding to sprinkler irrigation method, in combination with compost application, may be an interesting strategy to increase rice sustainability, increase the water and crop productivity, as well as reduce the risk of arsenic and cadmium accumulation in grains.

## Figures and Tables

**Figure 1 plants-12-00456-f001:**
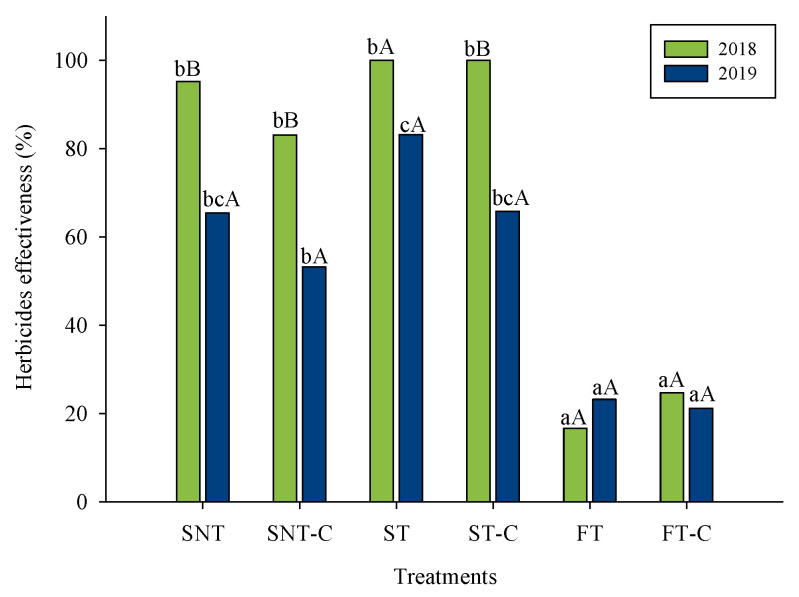
Effects of different management systems on herbicides effectiveness. Different letters indicate differences (*p* < 0.05) between treatments in the same year (lower case letters) and between years within the same treatment (upper case letters).

**Figure 2 plants-12-00456-f002:**
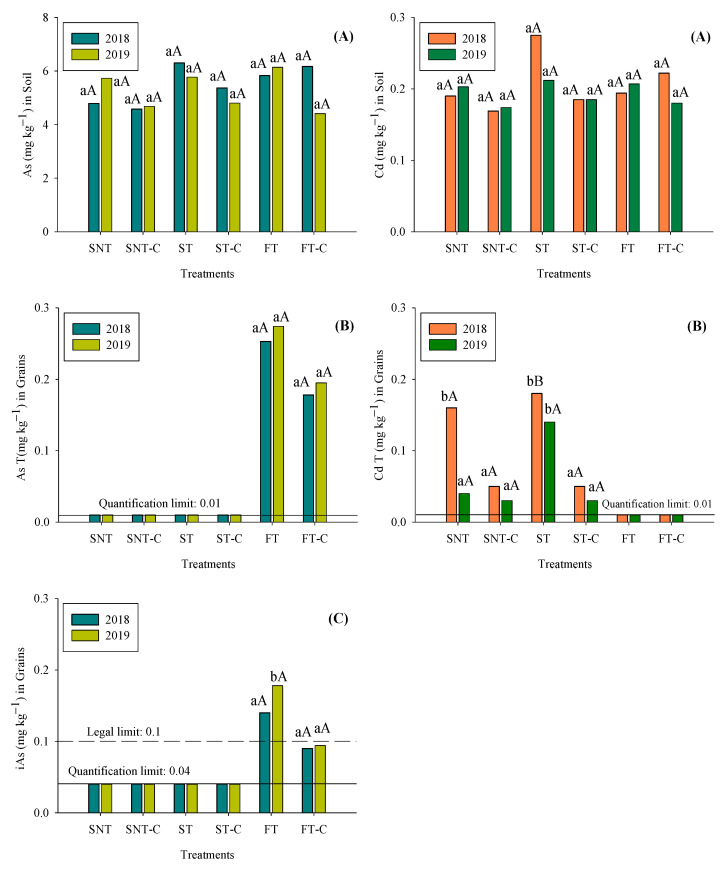
Medium-term effects of different management systems on concentrations of As and Cd in the soils (**A**), in the rice grains (**B**) and on concentrations of Inorganic As (iAs) in the rice grains (**C**). Different letters indicate differences (*p* < 0.05) between treatments in the same year (lower case letters) and between years within the same treatment (upper case letters). Note: the concentrations of organic fractions of As were always below the quantification limit (0.05 mg kg^−1^).

**Table 1 plants-12-00456-t001:** Medium-term effects of different management systems on rice yield components.

	GI (%)	RI (%)	Y (kg ha^−1^)	WP (g L^−1^)
2018				
SNT	56.5cA	87.0aB	9 226abA	1.07bA
SNT-C	51.9bcA	88.2aB	9 183abA	1.07bA
ST	52.2cA	88.6aB	10 307bA	1.20bA
ST-C	52.7cA	86.2aA	10 212bA	1.19bA
FT	46.2abB	88.4aB	8 343aA	0.513aA
FT-C	43.8aB	91.0aB	7 780aA	0.478aA
2019				
SNT	62.7bA	73.2aA	7 070aA	0.796bA
SNT-C	59.4bA	73.1aA	8 436aA	0.950bA
ST	62.9bB	82.3bA	11 625bA	1.31cA
ST-C	58.4bA	80.2bA	10 872bA	1.22cA
FT	31.6aA	74.7aA	7 921aA	0.516aA
FT-C	31.6aA	73.5aA	7 654aA	0.499aA
Y	NS	***	NS	NS
T	***	*	***	***
Y × T	*	NS	*	**

GI: Germination Index; RI: Ripening index; Y: Yield; WP: Water Productivity; ANOVA factors are Y: Year; T: Treatment; Y × T: Interaction Year * Treatment. F-values indicate the significance levels * *p* < 0.05; ** *p* < 0.01; *** *p* < 0.001, respectively, and NS: not significant. Different letters indicate differences (*p* < 0.05) between treatments in the same year (lower case letters) and between years within the same treatment (upper case letters).

## Data Availability

Not applicable.

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
