# Peer review of "Medium-Term Effects of Sprinkler Irrigation Combined with a Single Compost Application on Water and Rice Productivity and Food Safety"

_plants, 2023, doi:10.3390/plants12030456_

Round 1
Reviewer 1 Report
The manuscript presents a study on the medium-term effects of different irrigation and tillage methods combined with a single compost application on soil properties, water and rice productivity, as well as food safety, and demonstrates combination of sprinkler irrigation and compost application be the best strategy to rice management for the study area. The formal aspects of the paper are proper. The paper is well prepared and well-organized, it brings valuable results.
One small question for authors is that how you make sure the difference between weed control methods in two irrigation treatments wouldn’t affect the analysis result of weed control efficiency.
Please make sure all abbreviates use in your manuscript are clearly declared and consistent.
Line 29: :”between TF and SNT”, it’s not clear what TF means here. Also line 224-226: “NTS”, “TS”, and “TF” are not defined. Line 30-31: “with average savings of 55% of the total amount of water applied in FT.” is not clear here. Please clarify water saving in an independent sentence inside of a clause.
Line 107: change “development” to “develop”.
Line 176: sector “2.3. Data collection” should be “2.2. Data collection”. Change following sectors as well.
Line 244: guess “ AH” here should be “HA”.
Line 531: sector “2.4 Arsenic and cadmium …” should be “3.4 Arsenic and cadmium …”.
Line 614: sector “5. Conclusions” should be “4. Conclusions”.
Author Response
Reviewers' comments:
Reviewer#1
The manuscript presents a study on the medium-term effects of different irrigation and tillage methods combined with a single compost application on soil properties, water and rice productivity, as well as food safety, and demonstrates combination of sprinkler irrigation and compost application be the best strategy to rice management for the study area. The formal aspects of the paper are proper. The paper is well prepared and well-organized, it brings valuable results.
Thank you
One small question for authors is that how you make sure the difference between weed control methods in two irrigation treatments wouldn’t affect the analysis result of weed control efficiency.
Rainfall was not recorded prior and after 48 hours of herbicides application, which is sufficient time to prevent the potential washing of these compounds from the leaves as is suggested by the manufacturer. In addition, in order to avoid washing of herbicides, during this period, sprinkler treatments were not irrigated and flooded treatments were drained (before herbicides application).
Please make sure all abbreviates use in your manuscript are clearly declared and consistent.
As suggested by the Reviewer, all abbreviations have been defined and consistently used throughout the amended manuscript.
Line 29: :”between TF and SNT”, it’s not clear what TF means here. Also line 224-226: “NTS”, “TS”, and “TF” are not defined. Line 30-31: “with average savings of 55% of the total amount of water applied in FT.” is not clear here. Please clarify water saving in an independent sentence inside of a clause.
Thank you. The treatments abbreviations have been revised throughout the amended manuscript. As suggested by the Reviewer, water saving has been clarified in an independent sentence in the amended manuscript (abstract section).
Line 107: change “development” to “develop”.
As suggested by the Reviewer, “development” has been changed by “develop” in the amended manuscript.
Line 176: sector “2.3. Data collection” should be “2.2. Data collection”. Change following sectors as well.
Thank you. The numbers of subsections have been revised in the amended manuscript. Furthermore, as suggested by Reviewer #3, "Soil measurements" and "Properties of soil" subsections have been deleted in the amended manuscript. Therefore, the rest of subsections have been re-numbered.
Line 244: guess “ AH” here should be “HA”.
As suggested by the Reviewer #3, the properties of soil subsection has been deleted in the amended manuscript.
Line 531: sector “2.4 Arsenic and cadmium …” should be “3.4 Arsenic and cadmium …”.
Thank you. The Arsenic and cadmium subsection has been re-numbered in the amended manuscript.
Line 614: sector “5. Conclusions” should be “4. Conclusions”.
Thank you. “5. Conclusion” has been corrected by “4. Conclusions” in the amended manuscript
Best Regards,
David Peña Abades
Reviewer 2 Report
This paper presents a medium-term effects of sprinkler irrigation combined with a single compost application on water and rice productivity and food safety. It presents a change in the traditional management of rice irrigation, reducing water consumption, ensuring sustainability and food security.
The objectives of the work were generally achieved.
Specific comments:
The values presented must be represented according to international standards, in which the separation of significant number is by comma (,) and not period (.). Example line 143: “20,8%” and not “20.8%”. Replace in all paper.
Line 200, 2001 In the agronomic parameters, put how were measured the ripening index (RI), grain yield (Y), and water productivity (WP). Sánchez-Llerena et al. it's not open access, so readers don't have access.
Author Response
Reviewers' comments:
Reviewer#2
This paper presents a medium-term effects of sprinkler irrigation combined with a single compost application on water and rice productivity and food safety. It presents a change in the traditional management of rice irrigation, reducing water consumption, ensuring sustainability and food security.
The objectives of the work were generally achieved.
Specific comments:
The values presented must be represented according to international standards, in which the separation of significant number is by comma (,) and not period (.). Example line 143: “20,8%” and not “20.8%”. Replace in all paper.
The paper has been written in British English. In this language, while the comma is used as a thousand separator, the period is used as a decimal separator.
Line 200, 2001 In the agronomic parameters, put how were measured the ripening index (RI), grain yield (Y), and water productivity (WP). Sánchez-Llerena et al. it's not open access, so readers don't have access.
As suggested by the Reviewer, in the amended manuscript, it has been indicated how the RI, Y and WP were measured.
Best regards,
David Peña Abades
Reviewer 3 Report
Though the manuscript has scientific as well as practical soundness, the concept must be revised. I don't think that the soil properties can be assessed with the same concept as the crop parameters. I suggest to skip the soil related parts that could be analysed only on the basis of the reference data from 2015.
The text should be revised by a native English speaker.
I inserted several sticky notes in the text with my remarks, questions and corrections.

Author Response
Reviewers' comments:
Reviewer#3
Though the manuscript has scientific as well as practical soundness, the concept must be revised. I don't think that the soil properties can be assessed with the same concept as the crop parameters. I suggest to skip the soil related parts that could be analysed only on the basis of the reference data from 2015.
As suggested by the Reviewer (sticky note in the conclusion section), the sections of "Soil measurements" and "Properties of soil" have been deleted in the amended manuscript. Nonetheless, the physicochemical properties and enzymatic activities of soils have been included as Table S1 and Figure S1, respectively, in the supplementary material of the amended manuscript.
The text should be revised by a native English speaker.
As suggested by the Reviewer, the text has been revised by a native English speaker.
I inserted several sticky notes in the text with my remarks, questions and corrections.
The sticky notes have been revised and take into consideration in the amended manuscript (please see pdf file, plants-2142832-Reviewer#3).
Best regards,
David Peña Abades

Round 2
Reviewer 3 Report
The manuscript was significantly improved according to my suggestions. By deleting the confusing soil related parts, the text became more coherent. I agree with and accept that the data of the physicochemical properties and enzymatic activities of soils will be published in a table and a figure as supplementary materials.